# A Yeast-Based Model for Hereditary Motor and Sensory Neuropathies: A Simple System for Complex, Heterogeneous Diseases

**DOI:** 10.3390/ijms21124277

**Published:** 2020-06-16

**Authors:** Weronika Rzepnikowska, Joanna Kaminska, Dagmara Kabzińska, Katarzyna Binięda, Andrzej Kochański

**Affiliations:** 1Neuromuscular Unit, Mossakowski Medical Research Centre Polish Academy of Sciences, 02-106 Warsaw, Poland; wrzepnikowska@imdik.pan.pl (W.R.); dagkab@imdik.pan.pl (D.K.); kbinieda@imdik.pan.pl (K.B.); 2Institute of Biochemistry and Biophysics Polish Academy of Sciences, 02-106 Warsaw, Poland; kaminska@ibb.waw.pl

**Keywords:** Charcot-Marie-Tooth disease, neurodegenerative diseases, neuropathy, yeast model organism

## Abstract

Charcot–Marie–Tooth (CMT) disease encompasses a group of rare disorders that are characterized by similar clinical manifestations and a high genetic heterogeneity. Such excessive diversity presents many problems. Firstly, it makes a proper genetic diagnosis much more difficult and, even when using the most advanced tools, does not guarantee that the cause of the disease will be revealed. Secondly, the molecular mechanisms underlying the observed symptoms are extremely diverse and are probably different for most of the disease subtypes. Finally, there is no possibility of finding one efficient cure for all, or even the majority of CMT diseases. Every subtype of CMT needs an individual approach backed up by its own research field. Thus, it is little surprise that our knowledge of CMT disease as a whole is selective and therapeutic approaches are limited. There is an urgent need to develop new CMT models to fill the gaps. In this review, we discuss the advantages and disadvantages of yeast as a model system in which to study CMT diseases. We show how this single-cell organism may be used to discriminate between pathogenic variants, to uncover the mechanism of pathogenesis, and to discover new therapies for CMT disease.

## 1. Introduction

The peripheral neuropathies, also known as polyneuropathies, are a large group of disorders affecting the three types of peripheral nerves: motor, sensory, and autonomic. The clinical presentations of all neuropathies overlap, and the primary causes are numerous and varied. Infectious, immune-mediated, metabolic, toxic, vascular, genetic, and idiopathic forms can all be distinguished from one another. Hereditary neuropathies include Charcot–Marie–Tooth disease (CMT), also known as hereditary motor sensory neuropathy (HMSN); the hereditary motor neuropathies (HMN); the hereditary sensory and autonomic neuropathies (HSAN), also known as hereditary sensory neuropathy (HSN); and small fiber neuropathies (SFN). It is notable that the lines between one class and the next are relative and considered “blurred” or “fluid”. The CMT disease can be classified into several types/subtypes, depending on the mode of inheritance (dominant, recessive), the pattern of the injury (axonal, demyelinating) and the genes involved (more than 100 different genes have been identified so far). Despite this high heterogeneity, the clinical presentation allows for the “classical” CMT phenotype to be distinguished. Typically, the disease begins between the first and second decade of life, with weakening and wasting of the distal muscles, usually of the lower limbs, with accompanying sensory abnormalities. Some patients have skeletal deformities, the most common being pes cavus (a high arched foot). The muscle-wasting and weakness slowly progresses and worsens throughout the patient’s life. In addition to the “classical” CMT symptoms, the patient may also exhibit a wide range of additional symptoms, including hearing impairment, optic atrophy, vocal cord paresis, distal arthrogryposis, and even diaphragmatic weakness [1].

In the late 1960s, the overall prevalence of CMT in the population was estimated as being at the level of 1:2500 [2]. However, the most recent epidemiological studies have shown that there is considerable geographic variation, with a minimum prevalence in Serbia (9.7:100,000), and much higher levels in Norway (1:1250). In general, using different methodologies, the prevalence of CMT has been determined as ranging from 1:5000 to 1:10,000 in European populations [3]. Despite the relatively high prevalence of CMT in some populations, its subtypes belong to the group of rare or even ultra-rare diseases and, like most disorders in this group, suffer from the same problems, namely, verification of mutation pathogenicity, poorly understood molecular mechanisms and a lack of efficient treatments. In this review, we raise the issue of using yeast as a model for studying neuropathies, in particular CMT disease, and present how it may help to overcome these three problematic, but ultimately basic, issues. Yeast systems offer many advantages that are still poorly utilized to investigate neuropathies in general. Many researchers do not realize that yeast may be a convenient model for studying ongoing processes in peripheral nerves diseases. We present the huge potential of this simple, unicellular organism to improve diagnostics, expand the understanding of pathogenesis, and accelerate the development of treatment. The studies of CMT disorder using a yeast model included here have not been summarized in any review to date; hence, we hope that by showing the broad spectrum of possibilities that yeast systems present, it may be more widely adopted as a useful tool in CMT research.

## 2. Genetic Background of Charcot–Marie–Tooth Disease

CMT disease is characterized by an extreme genetic heterogeneity. To date, more than 1000 mutations have been described in more than 100 genes as causes of different CMT disease types (subtypes) [4]. Familial aggregation of CMT cases was identified at the beginning of the 20th century [5]. Around 70 years later, the first CMT-associated gene and the most common genetic cause of CMT was described (a 1.4 Mb tandem duplication located on chromosome 17p11.2-p12, encompassing the *PMP22* gene) [6,7]. The duplication of *PMP22*, responsible for CMT type 1A (CMT1A), is identified in more than 60% of CMT-affected patients [1]. Point mutations in three other genes—*GJB1*, *MFN2* and *MPZ* (causing CMTX1, CMT2, and CMT1B, respectively)—are responsible for around 30% of CMT cases [8]. At the opposite end of the spectrum are genes in which mutations have only been identified in isolated CMT families and lineages. This list includes mutations in *PRPS1, ATP7A*, and *IGHMBP2.*

Surprisingly, even in the era of next-generation sequencing (NGS), using whole exome sequencing (WES) technology, which enables for testing of all known CMT-causing mutations in a single approach, results in diagnosis for only 45% of CMT disease cases [1]. The relatively low ratio of positively verified CMT cases may be associated with structural variations that are not detectable in routine molecular analysis, expanding the mutation spectrum of existing CMT variants. These structural variations encompass both large translocations (e.g., a 1.35 Mb inversion on chromosome 7q36.3) and relatively small duplications (e.g., 118 kb, 6.25 kb) [9]. However, the list of genes responsible for CMT disease seems to be far from complete. In recent years, additional mutations in genes previously unassociated with CMT disease have been reported in single families [4]. For these small sample groups, the mechanism of pathogenicity associated with these genetic mutations remains unclear. Thus, despite access to powerful genetic tools, such as WES or whole genome sequencing (WGS), a correct and clear diagnosis based on a comprehensive genetic analysis is still not available for many patients.

The distinction between pathogenic and non-pathogenic variants of a disease is a problem for all genetic diseases. Solving this problem is important in order to assess the risk of disease in individual patients and for the development of therapies. The relationship between the gene and the disease can be established by comparing the frequency of rare variants in the patient group compared to the healthy population [10,11]. There are also various computational and experimental methods used to deduce the pathogenicity of rare variants. However, computational methods have the limited predictive potential [12,13,14,15,16,17], and the experimental evaluation of variant function in human cells is difficult, due to inefficient allele replacement methods and the presence of paralogs with overlapping functions. This makes tests in "humanized" model organisms, such as yeast, an attractive alternative.

## 3. Therapeutic Approaches for Charcot-Marie-Tooth Disorder

The vast majority of experimental therapies are dedicated to the most common CMT subtypes, i.e., CMT1A (*PMP22* gene duplication), CMTX1 (*GJB1* gene point mutations), CMT1B (*MPZ* gene mutations) and CMT2A (*MFN2* gene mutations), which, together, make up more than 90% of genetically confirmed CMT cases [18]. However, even in these cases, the proposed therapies are not universal. They usually target a single disease mechanism, and are thus dedicated to a specific set of mutations in the associated gene. Gene therapy has also been attempted as a treatment for CMT disease caused by less common mutations, including *IGHMBP2* and *SH3TC2* [19,20,21]. As the treatment strategies for different types of CMT were described in detail elsewhere [22], below, we summarize the most important information about possible therapies for CMT disorder.

CMT1A, the most common CMT disease subtype, is the result of elevated expression of *PMP22*. Down-regulation of *PMP22* is, therefore, the main therapeutic strategy targeting this disease subtype. Unfortunately, the first attempts to down-regulate *PMP22* gene expression in a mouse model (ascorbic acid) and in a transgenic rat model (progesterone receptor antagonists) could not be translated into human clinical trials [23,24,25]. The discovery that a combination of three medications already on the market (baclofen, sorbitol and naltrexone—PXT3003) was able to ameliorate the long-term phenotypical manifestation of peripheral neuropathy in a CMT1A rat model was a significant step forward (reviewed in [26]). Recently, encouraging news in relation to the Phase III trial of PXT3003 was reported [27]. Another promising pre-clinical therapeutic for patients carrying the CMT1A mutation is ADX71441 (a positive allosteric modulator of GABA_B_ receptors) [28], which was approved for phase I clinical trials for other diseases [29]. In addition, the use of antisense oligonucleotides (ASOs) appears to be a good strategy for the suppression of *PMP22* mRNA levels [30]. Nevertheless, it is notable that a variety of missense (amino acid substitutions), nonsense (premature termination due to stop codons), and frameshift mutations have also been described in *PMP22*, some of them causing more severe phenotypes. These mutations result in a different, not yet fully elucidated, pathogenic mechanism [31] that will require different therapeutic strategies.

The second most common form of CMT disease, CMTX1, is caused by mutations within the *GJB1* gene, and seems to be an optimal candidate for gene replacement therapy, due to the small size of the *GJB1* gene and the loss-of-function observed for the majority of *GJB1* mutations [32,33]. The treatment strategy for the next most common type of CMT, CMT1B, which is caused by mutations in *MPZ*, is mainly focused on relieving the effects of accumulated mutant forms of the protein in the endoplasmic reticulum (ER). However, this mechanism is not observed in all *MPZ* mutations, and some may manifest in different manners [31]. It was shown that curcumin was able to release MPZ mutant proteins from the ER to the cytoplasm and cause a significant decrease in apoptosis in HeLa cells [34]. Curcumin also demonstrated positive results in a CMT1B mouse model [35,36]. Sephin1 (a selective inhibitor of a holophosphatase), which attenuates stress resulting from misfolded proteins, also had a beneficial effect [37]. Finally, for CMT2A disease caused by mutations in the *MFN2* gene (encoding Mitofusin 2), abnormal mitochondrial trafficking has been reported for at least some mutations. In recent studies, mitofusin agonists have been shown to normalize mitochondrial pathology in the sciatic nerves of MFN2 Thr105Met mice [38].

Despite numerous attempts, no single effective therapeutic for CMT disorders has been registered on the market to date. The vast majority of CMT subtypes and mutations in “common” CMT genes with no “classical” (or an unknown) pathogenic mechanism have not been the subject of research for therapeutic approaches [39]. This state is a result of major barriers to research—including diverse molecular mechanisms for most CMT subtypes (even for different mutations in the same gene)—a lack of knowledge relating to the pathophysiology of specific variants, a deficiency of good disease models, and problems with translating results from animal models into humans. In this context, new models with which to identify new drugs may help to fill the gaps in available CMT therapies.

## 4. Studies of CMT in Yeast-Based Models for Human Genes with Yeast Orthologs

The simplicity of yeast is both its principle advantage and disadvantage as a model system. On the one hand, yeast provides an easy, cheap, and rapid platform for conducting research. On the other hand, are we truly able to use it to study processes in very complex systems such as neurons? Can yeast really help to solve the problems that neurogenetics faces: the unknown pathogenicity of rare sequence variants; unclear disease mechanisms; and a lack of effective therapies for patients? It has been shown that yeast can help in all of these cases. In this section, we will present the commonalities between a single-celled yeast and very sophisticated neurons, and how we can exploit these.

Although yeast and humans are separated by a billion years of evolution, a pairwise comparison of genes between these two species reveals more than 2041 groups of orthologs, representing 2386 yeast genes and 3673 human genes [40]. Moreover, there are more than 1000 functional complementation pairs, where a gene from one species can functionally replace (complement) its ortholog in the other [41]. This clearly indicates a significant conservation of function between such distant species, which opens a great number of research possibilities to explore. In the past 30 years, more than 400 yeast genes have been used to study their human counterparts [42]. Whether a human gene will complement mutations of its yeast ortholog cannot be confidently predicted based on the sequence alone [43]. Data related to experimental cross-species complementation are collected and are easy to extract from an open database (for details see [41]). The broad spectrum of possibilities of how to create and utilize yeast as a human model for diseases has been described elsewhere [42,44]. Here, we would like to highlight efficient neuropathy yeast-based models.

The CMT disease consists of a group of disorders displaying high genetic heterogeneity. If we looked at the list of genes associated with neuropathies (Table 1 and Table 2), with a particular focus on the function of proteins that they encode, we notice that these proteins occur in a diverse range of cellular pathways and processes. This ranges from the most common of processes, for example, translation (aminoacyl-transfer RNA (tRNA) synthetases), to highly specialized processes such as myelin sheath formation. It is not possible to study the formation of myelin using yeast, but it may be a good model to describe pathogenic mechanisms in more basic processes.

Looking at Table 1 and Table 2, of the more than 170 genes involved in various neuropathies, 60 have orthologs in yeast cells (Table 1 and Figure 1).

Some of these genes are so well conserved that even human genes may, at least in part, complement a lack of native yeast orthologs (Table 1). This gives a wide range of possibilities to model and study the neuropathy-associated mutations in yeast cells, which has advantages over other more complex models in terms of its low cost, growth rate, and genetic tractability. However, this opportunity is usually not fully exploited. In most cases, yeast is used only sporadically and to investigate one narrow aspect of a disease-associated mutation. Meanwhile, similar to other rare disorders, yeast-based neuropathy models may, at least partially, solve three of the major problems faced by researchers (Figure 2).

The first problem is the significance of rare sequence variants found in patients. As mentioned above, in the current era of NGS, several rare alleles are often identified in individual patients, but their impact on human health is usually poorly understood. This is one of the reasons for prolonged and incomplete diagnoses. There is an urgent need to develop a fast, cheap and reproducible system to test the pathogenicity of identified sequence variants. Yeast has been presented as a good platform for the study of aminoacylation activity, and, thus, the possible pathogenicity of human aminoacyl-transfer RNA (tRNA) synthetases (aaRSs) encoded by *ARS* genes [68]. AaRSs are key enzymes that catalyze the first reaction in protein biosynthesis, charging tRNAs with their cognate amino acids. To date, mutations in six *ARS* genes have been associated with CMT disorders (*GARS*, *YARS*, *AARS*, *MARS*, *HARS*, and *WARS,* encoding glycyl- tyrosyl- alanyl-, methionyl-, histidyl-, tryptophanyl-RS, respectively) [53,69,70,71,72,73]. It has been shown that human orthologs can complement the lethality of the deletion of the alanyl-, glycyl-, histidyl-, tyrosyl-RS genes (*ALA1*, *GRS1*, *HTS1*, *TYS1*, respectively) in yeasts and restore the growth of these cells (see Table 1). The other two human *ARS* genes associated with CMT (*MARS* and *WARS*) possess yeast orthologs (*MES1* and *WRS1*, respectively), which allows researchers to model the mutations found in patients in the corresponding yeast genes. Testing newly identified rare variants in yeast not only allows the identification of loss-of-function (functional null; hypomorphic) alleles but can also reveal gain-of-function (i.e., hypermorphic) alleles [45,73]. Even though a human full-length wild-type *WARS* failed to complement a *wrs1* deficiency in yeast cells, a mutant *WARS* incorporating a H257R substitution could partially complement the lack of *WRS1*, which implies that H257R substitution may change the structure of human tryptophanyl-RS [73].

Another example of using a yeast-based system to investigate rare sequence variants is the study of mutations in the *POLG* gene, which encodes the catalytic subunit of mtDNA polymerase γ. Pathological mutations in this gene are usually associated with severe mitochondrial disorders, but may also manifest as an isolated neuropathy [74]. Yeast is a suitable model organism for the study of alleles resulting in severe oxygen and/or respiration impairment and mitochondrial dysfunction, due to its ability to survive without oxidative phosphorylation. Comparable phenotypes (the harmful effects observed in yeast reproduce the severity of the phenotypes in humans), the possibility to study variants in heteroallelic states, and easy and fast analysis make budding yeast an excellent model for testing mutations in *POLG*. In *POLG*-associated diseases, yeast has enabled researchers to distinguish pathogenic mutations from other single-nucleotide polymorphisms; to show that some polymorphisms may act as phenotypic modifiers; and has demonstrated that certain mutations are not the only cause of a pathology, highlighting the need for further genetic analysis [75,76,77,78,79,80,81].

The second problem is a deficiency or complete lack of knowledge related to the molecular mechanisms underlying the pathogenicity of mutations. Information about the cellular processes that trigger pathogenic changes and ultimately lead to the symptoms observed in patients may provide clues for the production of effective therapeutics. It may also reveal the possible toxicity of the drugs used and which drugs should be avoided when treating specific patients. It may also enable recommendations to be made for lifestyle changes to improve the quality of life and slow the disease’s progression. An example of using yeast to learn about the mechanism of pathogenicity of a specific mutation is the study of the human gene *MFN2* encoding Mitofusin 2. Mutations in *MFN2* result in the most frequent axonal form of CMT (CMT2A) [82]. Mitofusin 2 represents a key player in mitochondrial fusion, trafficking, turnover and the formation of contacts with other organelles [83]. Yeast mitofusin, Fzo1, was used to study the consequences of CMT2A mutations associated with the human *MFN2* gene. It was shown that one mutation in particular (causing a substitution analogous to the I213T substitution in Mfn2) is highly deleterious for protein function and stability. Other mutations had variable effects, causing either no phenotype, or a subtle alteration of mitochondrial morphology [84]. The study of Mfn2 function is also of importance because mitofusins belong to the group of proteins important for the formation, regulation, and function of endoplasmic reticulum (ER) and mitochondrial membrane contact sites (MCSs), called mitochondria-associated membranes (MAMs). MCSs are structures where membranes of different organelles are close and connected by a proteinaceous tether but do not fuse. The genes affecting homeostasis in MAMs are over-represented in the group of genes causing several hereditary neurodegenerative disorders, such as Alzheimer’s disease [85,86,87,88] and Chorea-acanthocythosis [89]. This is because MCSs are involved in various processes, including mitochondrial dynamics (fusion, fission), lipids metabolism, autophagy, cell survival, energy metabolism, calcium homeostasis, and protein folding [90,91,92]. In the group of genes causing CMT disorders, besides Mfn2, there are other proteins with and without orthologs in yeast, which take part in processes at MAMs, such as VAPB, Opa1, or GDAP1 [93]. Studies on the function of one MCSs component may help uncover the role of MCSs in the development of several other neurodegenerative diseases. More details about the role of MAMs in neurodegeneration can be found in other reviews [85,86,87,88].

The third and final challenge, facing not only neuropathies but all rare diseases, is a lack of therapies. This problem was more specifically described in the Section 3. Yeast may also serve as a rapid and cheap platform with which to screen potentially active compounds, and for a detailed analysis of the cellular effects of drugs (see Section 6).

## 5. Studies of CMT for Human Genes Lacking Orthologs in the Yeast Model

The most simple yeast-based neuropathy models are based on the homology between human and yeast genes. However, even in the absence of clear orthologs, yeast may still be used to study the three problematic areas for rare diseases identified in Section 4.

The strategy that allows for this is based on the belief that yeast and the cells of higher organisms are built and function according to the same principles. Thus, human proteins may still be able to modulate essential cellular pathways in yeast. This rule has been shown in the case of yeast-based studies of neurodegenerative disorders including Alzheimer’s disease, Parkinson’s disease, and Huntington’s disease. Here, the heterologous expression of pathological variants of aβ-peptide, α-synuclein or poly-Q repeats of different lengths in the yeast allowed researchers to dissect the molecular mechanisms underlying the pathology of mutant proteins, identify potential drug targets and select active compounds by screening the available drug libraries [94,95,96,97,98]. Similarly, in the case of genes whose mutations lead to CMT diseases and which have no orthologs (Table 2), their expression in yeast may still affect cells and limit their growth.

This strategy, testing the activity of a human protein without an ortholog in yeast, has already been used to study neuropathy associated with mutations in the *GDAP1* gene encoding ganglyoside induced differentiation associated protein 1. Although the production of this protein does not affect the growth of wild-type yeast cells, it has manifestations at the molecular level, as shown in two different studies [99,100]. In both studies, the effect of heterologous *GDAP1* expression in the wild-type and different yeast mutants was tested. Based on the observation that increased expression of *GDAP1* in COS-7 cells results in mitochondrial fragmentation, which is probably due to interference with the mechanism of mitochondrial fission, the influence on yeast mitochondria morphology was investigated [99,100]. The expression of *GDAP1* either did not affect yeast mitochondria morphology [99], or caused increased fragmentation of the mitochondria, depending on the specific experimental procedures [100], and also changed mtDNA maintenance [100]. *GDAP1* expression in the *fis1Δ* mutant, defective for mitochondrial fission and for G2/M progression during the cell cycle, did not eliminate the mitochondrial fission defect, but reversed the cell cycle delay phenotype of this mutant [99]. Although the results obtained did not define the molecular function of the GDAP1 protein, they indicate the pathway for further research. Moreover, finding a clear phenotype allowed for further testing of the effect of pathogenic *GDAP1* gene missense variants. Since all the investigated variants did not reverse the cell cycle delay phenotype of a *fis1Δ* mutant, it suggests that this phenotype can be used to study new *GDAP1* variants and distinguish pathogenic from non-pathogenic alleles identified in patients during the diagnostic process. However, it is not possible to differentiate between variants in terms of the strength of the clinical symptoms induced.

A more functional system with which to evaluate the pathogenicity of *GDAP1* gene mutations was reported in our previous study [100]. In this case, the *csg2Δ* mutant was used with the deletion of a gene required for mannosylation of inositolphosphorylceramide. The sensitivities of the *csg2Δ* strain to stress conditions, the presence of tunicamycin and high concentrations of calcium ions (Ca^2+^) were all suppressed by the expression of the wild-type *GDAP1* allele, but not by *GDAP1* variants encoding proteins that lost the ability to correctly localize to the mitochondria. Different *GDAP1* variants (point mutants) exhibited differing abilities to suppress these phenotypes. This system, in addition to enabling testing of the pathogenicity of *GDAP1* alleles isolated from patients, should also allow for their classification in terms of the severity of the resulting clinical phenotypes.

These two working systems show that GDAP1 functionality can be assessed in yeast cells even though yeast has no functional orthologs. This is due to the fact that they participate in conserved cellular processes (Table 2) and, consequently, there are functional orthologs of their partner proteins (Figure 2). Based on this principle, it is possible to build additional yeast-based models for other genes that are involved in CMT disease, but do not have yeast orthologs. An example of this is the human gene *MTMR2* coding for Myotubularin 2-related protein, a member of the myotubularin family of phosphoinositide lipid phosphatases. Myotubularin 2 functionally interacts with the phosphoinositide 5-phosphatase protein FIG4, amongst other proteins in Schwann cells and in neurons [101]. Mutations in the *FIG4* gene, similar to in *MTRM2*, are described as resulting in CMT disease. The human FIG4 has a yeast ortholog, making it is possible to build a model to directly study *FIG4*, and to indirectly test *MTMR2* function in a yeast model.

Another possible use of the preservation of biochemical pathways is associated with the CMT subtype, caused by mutations in the *MPZ* gene. Some mutant MPZ proteins are retained in the ER, where their accumulation triggers an induction of the unfolded protein response (UPR). Although the downregulation of UPR has already been shown to have a positive effect in CMT1B, the problem associated with distinguishing between pathogenic and non-pathogenic variants and the subsequent search for mechanisms of pathogenicity remains. In a case of mutations in the *MPZ* gene, a yeast model can be used because of the conservation of biochemical pathways.

As a result of UPR induction in mammalian cells, the eIF2 kinases PERK and GCN2 are activated and phosphorylate the translation initiation factor 2 (eIF2a). This leads to the synthesis of transcription factor ATF4 and increased production of the CHOP protein. The induction of this response has been shown to cause demyelination in CMT1B. Depletion of CHOP or its subsequent target, Gadd34, improves myelination [102]. The mammalian signaling pathway leading to eIF2a phosphorylation is homologous to the well-studied general control response in yeast, in which phosphorylation of eIF2a activates genes involved in amino acid biosynthesis. Thus, mammalian cells use a conserved pathway to regulate gene expression in response to various stresses [103]. Therefore, it is possible to use yeast to study the pathogenicity of mutations by monitoring the effects caused by the presence of MPZ protein variants at the molecular level.

Finally, with models such as those described above, we can study the pathogenicity of variants and mechanisms of pathogenesis and use them to find therapies by searching libraries of small molecules. This experimental approach could and should be far more widely applied to the investigation of CMT diseases.

## 6. Repositioning of Drugs in Hereditary Neuropathies

In the case of common disorders, effective medicines can be found and tested in clinical trials involving thousands of patients, but for rare and ultra-rare diseases, the classical approaches to drug discovery are very difficult to follow. This is due to the small number of patients and the lack of economic justification for pharmaceutical companies to engage large resources in research that will not be profitable for them.

This problem applies to CMT disease, which can be classified as a rare or even ultra-rare disease. In such cases, one favorable solution is a drug repurposing strategy. Drug repurposing (also known as repositioning, reprofiling, rediscovering or redirecting) may be defined as developing new uses for a drug beyond its original intended use or initially approved use. The best example of drug repurposing is that of chlorpromazine. In 1950, chlorpromazine, synthesized as a potential antimalarial drug, was administered to patients before surgery. Due to its unexpected sedative effects, a weak anti-malarial drug has become a powerful medicine used in both psychiatry (acute mania) and neurology (chorea, epilepsy, muscle spasms, etc.) [104]. Drug repositioning radically reduces the cost of clinical trials, prevents the withdrawal of a drug from the market due to low interest, and usually allows well-known and cheap pharmaceuticals to be selected as the preferred option. This latter feature is especially important when developing therapies for rare disorders, as the alternatives (e.g., gene therapy or cell therapy) are extremely expensive.

Yeast models have great potential to be used as a platform for the screening of drugs libraries to obtain preliminary results. Such models benefit from a fast turnaround, low cost, and easy testing. Yeast models have been successfully used to search for active compounds against mitochondrial diseases [105,106,107], central nervous system diseases [108], and copper-deficiency disorders [109]. Thus, it seems reasonable that, for peripheral neuropathies, yeast models may also be applied and allow for drug repositioning. Nearly a third of human genes involved in the pathogenesis of CMT have yeast orthologs, making it relatively easy to create yeast-based models. For the other cases, there is the possibility of finding a convenient, easy to quantify phenotype (e.g., a toxic effect associated with the expression of gene variants), which may be used for drug screening. The other option is the use of phenologs, which are the phenotype-level equivalent of gene orthologs. Two sets of deeply conserved genes between extremely evolutionary divergent organisms, i.e., yeast and humans, may be manifested in the different molecular contexts as two dissimilar phenotypes. For example, cell-wall maintenance in yeast and vascular growth in human are phenologous processes: they share the same genes conserved between yeast and human. The failure of the equivalent genes will lead to disturbed angiogenesis in humans and reduced cell-wall maintenance in yeast. Using this logic, thiobendazole—marketed as an antifungal drug—would also act as an angiogenesis inhibitor in humans [110,111]. Similarly, phenologous processes may also be used for drug selection in CMT disease.

The repurposing strategy may be key to finding therapies, especially for very rare subtypes of CMT. To date, this approach has been poorly used for CMT diseases. Only a few CMT genes and their mutations (*AARS, ATP7A, GARS, HARS, HINT1*) have been modeled in yeast so far, and any have not been used for drug repurposing, despite the systems being ready to test collections of highly bioavailable drugs of known toxicity in models to confirm their action and to select the effective dose.

## 7. Outlook

Rapidly developing technologies provide ever-improving diagnostics, pathology monitoring and therapies. However, they also reveal the limitations of current approaches and “black holes” in our knowledge. Returning to more basic, simple models may be a key for further research regarding different, and especially very rare, neuropathies. Included in this work are examples that clearly indicate that yeast models have a great potential to serve as neuropathic models. These models are currently underused. Yeast models are a promising tool for the determination of the pathogenicity of newly discovered rare sequence variants and their influence on the progression of a disease. In addition, they may also provide an excellent platform for studying the molecular and cellular background of pathogenesis and offer a cheap, easy, and fast system for the high-throughput screening of genetic and chemical suppressors that would give rise to potential therapeutics. Yeast models have advantages over other models in terms of cost, ease of growth and use, and the facilities necessary to implement them. They could be used as a convenient tool in almost all laboratories, and we hope that they will continue to be developed to support the study of more hereditary neuropathies.

## Figures and Tables

**Figure 1 ijms-21-04277-f001:**
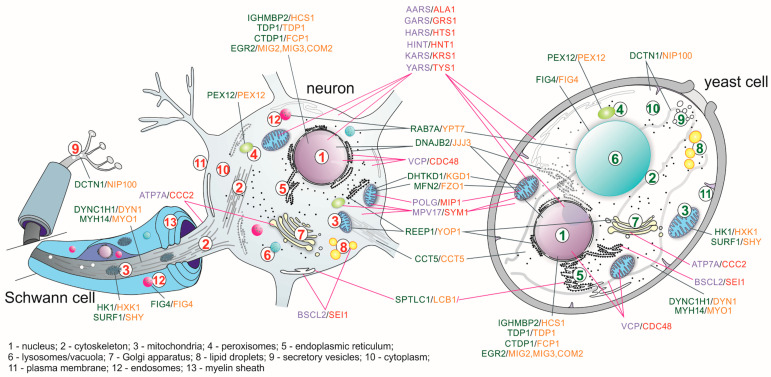
Scheme showing the localization of selected proteins, in which mutations have been associated with hereditary peripheral neuropathies in a human nerve cell and a yeast *Saccharomyces cerevisiae* cell. The violet color indicates human proteins complementing mutations in yeast proteins, marked in red; green indicates human proteins possessing orthologs in yeast *S. cerevisiae* that are marked orange.

**Figure 2 ijms-21-04277-f002:**
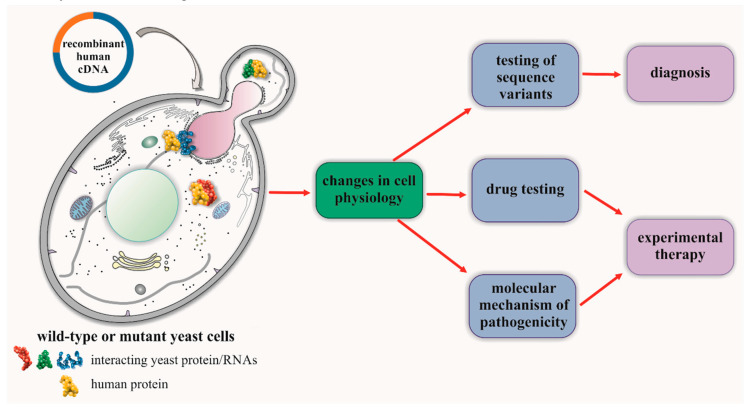
Yeast as a system to evaluate functional effects of human genetic variations. The coding part of a human gene (cDNA) is inserted under a yeast regulatory sequence and transformed into yeast cells, where it is expressed. The resulting protein interacts with yeast cellular components (proteins, RNAs, lipids, etc.) and affects the cell physiology, leading to the selected phenotypes which may be monitored. The system presented may be used to test unknown sequence variants to improve diagnosis, or for screening drug candidates and investigating the molecular mechanisms of pathogenicity to develop future experimental therapies.

**Table 1 ijms-21-04277-t001:** Human Genes Associated with Neuropathies and Their Yeast Orthologs.

Genes Complementing Yeast *S. cerevisiae* Orthologs Mutation				
Human Gene	Yeast Gene	Function of the Protein	Comments	Source
*AARS*	*ALA1*	Alanyl-tRNA synthetase	Wild-type *AARS* improved some yeast growth at 30 °C but more robustly at 37 °C [45]	[46]
*ATP7A*	*CCC2*	Copper-transporting P-type ATPase		[47]
*BSCL2*	*SEI1*	Seipin: necessary for correct lipid storage and lipid droplets maintenance	Complements the defects in lipid droplets in *sei1*Δ strain	[48]
*COX10*	*COX10*	Heme A:farnesyltransferase; functions in the maturation of the heme A, a prosthetic group of COX complex		[49,50]
*FXN*	*YFH1*	Frataxin: a component of a multiprotein complex that assembles iron–sulfur (Fe–S) clusters in the mitochondrial matrix		[51]
*GARS*	*GRS1*	Glycyl-tRNA synthetase		[52]
*HARS*	*HTS1*	Histidyl-tRNA synthetase		[53]
*HINT1*	*HNT1*	Hydrolyzes purine nucleotide phosphoramidates with a single phosphate group		[54]
*HMBS*	*HEM3*	Hydroxymethylbilane synthase: the third enzyme of the heme biosynthetic pathway		[55,56]
*MPV17*	*SYM1*	An inner-membrane mitochondrial protein; may form a channel in the inner mitochondrial membrane, supplying the matrix with desoxynucleotide phosphates and/or nucleotide precursors		[57]
*OPA1*	*MGM1*	Dynamin-related GTPase that is essential for normal mitochondrial morphology by regulating the mitochondrial fusion	*OPA1* cannot substitute the *MGM1* gene; however, chimeric protein composed of the N-terminal region of Mgm1 fused with the catalytic region of OPA1 is able to complement the *mgm1* null mutant	[58]
*POLG*	*MIP1*	Mitochondrial DNA polymerase gamma	The yeast mitochondrial localization signal was retained	[59]
*VAPB*	*SCS22 SCS2*	A type IV membrane protein found in plasma and intracellular vesicle membranes	Expression of *VAPB* partially compensated for the inositol auxotrophy *scs2*Δ*scs22*Δ yeast strain	[60]
*VCP*	*CDC48*	A member of the AAA ATPase family of proteins; plays a role in protein degradation, intracellular membrane fusion, DNA repair and replication, regulation of the cell cycle, and activation of the NF-kappa B pathway	Wild type VCP partially suppressed the temperature sensitivity growth of *cdc48-3* but not the cold sensitivity growth of *cdc48-1* and null mutation	[61]
*YARS*	*TYS1*	Tyrosyl-tRNA synthetase		[62]
**Genes Possessing Orthologs in Yeast *S. cerevisiae***				
**Human Gene**	**Yeast Gene**	**Protein Function**		
*ABCA1*	*YOL075C*	A member of the superfamily of ATP-binding cassette (ABC) transporters		
*AIMP1*	*ARC1*	A multifunctional polypeptide with both cytokine and tRNA-binding activities		
*ATP1A1*	*ENA5* *ENA1* *ENA2*	Catalytic component of the active enzyme, which catalyzes the hydrolysis of ATP coupled with the exchange of sodium and potassium ions across the plasma membrane		
*C12ORF65*	*RSO55*	A mitochondrial matrix protein that appears to contribute to peptide chain termination in the mitochondrial translation machinery		
*CCT5*	*CCT5*	A molecular chaperone that is a member of the chaperonin containing TCP1 complex (CCT), also known as the TCP1 ring complex (TRiC)		
*CHCHD10*	*MIX17*	A mitochondrial protein that is enriched at cristae junctions in the intermembrane space; it may play a role in cristae morphology maintenance or oxidative phosphorylation		
*CLP1*	*CLP1*	A member of the Clp1 family; it is a multifunctional kinase which is a component of the tRNA splicing endonuclease complex and a component of the pre-mRNA cleavage complex II		
*CLTCL1*	*CHC1*	The clathrin heavy chain protein		
*COX6A1*	*COX13*	Cytochrome C oxidase subunit		
*CTDP1*	*FCP1*	A protein which interacts with the carboxy-terminus of the RAP74 subunit of transcription initiation factor TFIIF, and functions as a phosphatase that dephosphorylates the C-terminus of POLR2A (a subunit of RNA polymerase II), making it available for initiation of gene expression		
*DCTN1*	*NIP100*	The largest subunit of dynactin		
*DHTKD1*	*KGD1*	A component of a mitochondrial 2-oxoglutarate-dehydrogenase-complex-like protein involved in the degradation pathways of several amino acids		
*DNAJB2*	*JJJ3*	Almost exclusively expressed in the brain, mainly in the neuronal layers; encodes a protein that shows sequence similarity to bacterial DnaJ protein and the yeast ortholog		
*DYNC1H1*	*DYN1*	Dynein cytoplasmic heavy chain; dyneins are a group of microtubule-activated ATPases that function as molecular motors		
*EGR2*	*MIG2; MIG3; COM2*	A transcription factor		
*EXOSC3*	*RRP40*	Non-catalytic component of the human exosome		
*EXOSC8*	*RRP43*	A 3’-5’ exoribonuclease that specifically interacts with mRNAs containing AU-rich elements		
*FIG4*	*FIG4*	Phosphoinositide 5-phosphatase		
*GMPPA*	*PSA1*	GDP-mannose pyrophosphorylase		
*HK1*	*HXK1* *HXK2* *GLK1* *EMI2*	Hexokinase 1		
*IGHMBP2*	*HCS1*	Helicase superfamily member that binds a specific DNA sequence from the immunoglobulin mu chain switch region		
*ELP1 (IKBKAP)*	*IKI3*	Scaffold protein and a regulator for three different kinases involved in proinflammatory signaling		
*MARS*	*MES1*	Methionyl-tRNA synthetase		
*MCM3AP*	*SAC3*	Involved in the export of mRNAs to the cytoplasm through the nuclear pores, promoting somatic hypermutations		
*MFN2*	*FZO1*	Mitofusin: participates in mitochondrial fusion		
*MT-ATP6*	*ATP6*	Mitochondrial membrane ATP synthase		
*MYH14*	*MYO1*	Member of the myosin superfamily		
*PDHA1*	*PDA1*	Pyruvate dehydrogenase subunit		
*PDK3*	*PKP1*	One of the three pyruvate dehydrogenase kinases that inhibits the PDH complex by phosphorylation of the E1 alpha subunit		
*PEX12*	*PEX12*	Belongs to the peroxin-12 family, proteins that are essential for the assembly of functional peroxisomes		
*PNKP*	*HNT3*	Involved in DNA repair		
*PRPS1*	*PRS4 PRS2 PRS3*	Enzyme that catalyzes the phosphoribosylation of ribose 5-phosphate to 5-phosphoribosyl-1-pyrophosphate, which is necessary for purine metabolism and nucleotide biosynthesis		
*RAB7A*	*YPT7*	RAB family members, regulate vesicle traffic in the late endosomes and also from late endosomes to lysosomes		
*REEP1*	*YOP1*	Mitochondrial protein that functions to enhance the cell surface expression of odorant receptors		
*SCO2*	*SCO1*	One of the COX assembly factors		
*SEPT9 (SEPTIN9)*	*CDC10 CDC3*	Member of the septin family involved in cytokinesis and cell cycle control		
*SETX*	*SEN1*	Contains a DNA/RNA helicase domain at its C-terminal end which suggests that it may be involved in both DNA and RNA processing		
*SIGMAR1*	*ERG2*	Receptor protein that interacts with a variety of psychotomimetic drugs, including cocaine and amphetamines		
*SLC25A19*	*TPC1*	Mitochondrial transporter mediating uptake of thiamine pyrophosphate into mitochondria		
*SPTLC1*	*LCB1*	The long chain base subunit 1 of serine palmitoyltransferase		
*SPTLC2*	*LCB2*	Subunit of serine palmitoyltransferase		
*SURF1*	*SHY1*	Localized to the inner mitochondrial membrane, involved in the biogenesis of the cytochrome c oxidase complex		
*TDP1*	*TDP1*	Is involved in repairing stalled topoisomerase I-DNA complexes by catalyzing the hydrolysis of the phosphodiester bond between the tyrosine residue of topoisomerase I and the 3-prime phosphate of DNA		
*UBA1*	*UBA1*	Catalyzes the first step in ubiquitin conjugation to mark cellular proteins for degradation		
*WARS*	*WRS1*	Tryptophanyl-tRNA synthetase		

List of genes was created based on [63]. Yeast orthologs were find using GeneCard [64] and YOGI databases [65]. Function of protein was described based on the GeneCard, UniProt, and OMIM databases [64,66,67].

**Table 2 ijms-21-04277-t002:** Human Genes Associated with Neuropathies with No Yeast Orthologs.

Process	Gene	Protein Function
Adhesion	*FBLN5*	Fibulin 5: extracellular matrix protein essential for elastic fiber formation; promotes adhesion of endothelial cells; may play a role in vascular development and remodeling
Apoptosis	*AIFM1*	Flavoprotein essential for nuclear disassembly in apoptotic cells, and found in the mitochondrial intermembrane space in healthy cells
Autophagy	*RETREG1* *(FAM134B)*	Endoplasmic reticulum-anchored autophagy receptor that mediates ER delivery into lysosomes through sequestration into autophagosomes
	*TECPR2*	Implicated in autophagy
Cytoskeleton organization	*DST*	Dystonin: cytoskeletal linker protein
	*FGD4*	Activates CDC42 by GDP/GTP exchange; binds to actin filaments; is involved in the regulation of the actin cytoskeleton and cell shape
	*GSN*	Gelsolin: calcium-regulated protein functions in both assembly and disassembly of actin filaments
	*INF2*	A member of the formin family: severs actin filaments and regulates their polymerization and depolymerization
	*MICAL1*	Monooxygenase that oxidizes methionine residues on actin, thereby promoting depolymerization of actin filaments
	*NEFH*	Neurofilament heavy polypeptide
	*NEFL*	Neurofilament light polypeptide
	*TUBB3*	A class III member of the beta tubulin protein family
Endoplasmic reticulum organization	*ATL1*	Alastin 1: GTPase functions in endoplasmic reticulum tubular network biogenesis
	*ATL2*	Atlastin 2: GTPase functions in formation of endoplasmic reticulum
	*ATL3*	Alastin 3: dynamin-like GTPase required for the proper formation of the endoplasmic reticulum tubules
	*ARL6IP1*	Transmembrane protein: plays a role in the formation and stabilization of endoplasmic reticulum tubules; negatively regulates apoptosis; regulates glutamate transport
Mitochondria functioning	*TWNK* *(C10ORF2)*	DNA helicase: involved in mitochondrial DNA (mtDNA) metabolism
	*GDAP1*	Regulates mitochondrial morphology and transport; participates in calcium homeostasis; regulates redox state of cell
	*NDUFAF5*	Mitochondrial protein required for complex I assembly
	*SLC25A46*	Functions in promoting mitochondrial fission, and prevents the formation of hyperfilamentous mitochondria
Myelination	*ARHGEF10*	A Rho guanine nucleotide exchange factor (GEF)
	*CNTNAP1*	Required for radial and longitudinal organization of myelinated axons
	*DRP2*	Dystrophin-related protein 2: required for normal myelination and for normal organization of the cytoplasm and the formation of Cajal bands in myelinating Schwann cells
	*FAM126A*	Hyccin: Component of a complex regulating phosphatidylinositol 4-phosphate; may play a part in the beta-catenin/Lef signaling pathway
	*MPZ*	Specifically expressed in Schwann cells of the peripheral nervous system; a type I transmembrane glycoprotein that is a major structural protein of the peripheral myelin sheath
	*PLP1*	A transmembrane protein that is the predominant component of myelin
	*PMP2*	Localizes to myelin sheaths of the peripheral nervous system; is thought to provide stability to the sheath
	*PMP22*	An integral membrane protein that is a major component of myelin in the peripheral nervous system
	*PRX*	A protein involved in peripheral nerve myelin upkeep
	*SH3TC2*	Expressed in Schwann cells: interacts with the small guanosine triphosphatase Rab11, which is known to regulate the recycling of internalized membranes and receptors back to the cell surface
Lipid metabolism	*ABHD12*	Catalyzes the hydrolysis of 2-arachidonoyl glycerol (2-AG), the main endocannabinoid lipid transmitter that acts on cannabinoid receptors
	*ASAH1*	Acid ceramidase: a lysosomal protein that hydrolyzes sphingolipid ceramides
	*CYP27A1*	Sterol 26-hydroxylase: cytochrome P450 monooxygenase that catalyzes hydroxylation of cholesterol and its derivatives
	*DGAT2*	Diacylglycerol O-acyltransferase 2: one of two enzymes which catalyzes the final reaction in the synthesis of triglycerides
	*GALC*	Galactocerebrosidase: a lysosomal protein which hydrolyzes the galactose ester bonds of galactosylceramide, galactosylsphingosine, lactosylceramide, and monogalactosyldiglyceride
	*GLA*	Alpha-galactosidase A: hydrolyses the terminal alpha-D-galactosyl moieties from glycolipids and glycoproteins
	*HADHA*	The alpha subunit of the mitochondrial trifunctional protein, which catalyzes the last three steps of mitochondrial beta-oxidation of long chain fatty acids
	*HADHB*	The beta subunit of the mitochondrial trifunctional protein, which catalyzes the last three steps of mitochondrial beta-oxidation of long chain fatty acids
	*HEXA*	Beta-hexosaminidase subunit alpha: involved in degradation of GM2 gangliosides, and other molecules containing terminal N-acetyl hexosamines
	*MTMR2*	Member of the myotubularin family of phosphoinositide lipid phosphatases: possesses phosphatase activity towards phosphatidylinositol-3-phosphate and phosphatidylinositol-3,5-bisphosphate
	*PLA2G6*	A2 phospholipase
Protein processing	*BAG3*	Co-chaperone for HSP70 and HSC70 chaperone proteins: acts as a nucleotide-exchange factor (NEF) promoting the release of substrate
	*DCAF8*	Interacts with the Cul4-Ddb1 E3 ubiquitin-protein ligase complex; may function as a substrate receptor
	*DNAJC3*	Acts as a co-chaperone of BiP, a major endoplasmic reticulum-localized member of the HSP70 family of molecular chaperones that promote normal protein folding
	*FBXO38*	Substrate recognition component of a SCF (SKP1-CUL1-F-box protein) E3 ubiquitin-protein ligase complex
	*GAN*	Gigaxonin: plays a role in neurofilament architecture and is involved in mediating the ubiquitination and degradation of some proteins
	*HSPB1*	A member of the small heat shock protein (HSP20) family: plays a role in stress resistance and actin organization
	*HSPB3*	A member of the small heat shock protein (HSP20) family: inhibitor of actin polymerization
	*HSPB8*	Belongs to the superfamily of small heat-shock proteins (HSP20): displays temperature-dependent chaperone activity
	*KLHL13*	Functions as an adaptor protein of a BCR (BTB-CUL3-RBX1) E3 ubiquitin-protein ligase complex required for mitotic progression and cytokinesis
	*LRSAM1*	E3 ubiquitin-protein ligase: involved in various functions
	*MME*	Neprilysin: membrane metalloendopeptidase
	*RNF170*	RING domain-containing protein that resides in the endoplasmic reticulum (ER) membrane; functions as an E3 ubiquitin ligase and mediates ubiquitination and processing of inositol 1,4,5-trisphosphate (IP3) receptors via the ER-associated protein degradation pathway
	*SACS*	Sacsin: co-chaperone which acts as a regulator of the Hsp70 chaperone
	*SBF1*	Myotubularin-related protein: acts as an adapter for the phosphatase MTMR2; promotes the exchange of GDP to GTP
	*TRIM2*	Functions as an E3-ubiquitin ligase: plays a neuroprotective function
	*VRK1*	Serine/threonine-protein kinase
	*WNK1*	Serine/threonine kinase which plays an important role in the regulation of electrolyte homeostasis, cell signaling, survival, and proliferation
Signaling	*ADCY6*	Belongs to the adenylate cyclase family of enzymes responsible for the synthesis of cAMP
	*AHNAK2*	Nucleoprotein: may play a role in calcium signaling
	*DHH*	Signaling molecules that play an important role in regulating morphogenesis
	*GJB1*	Gap junction beta-1 protein: a member of the gap junction protein family
	*GJB3*	Gap junction beta-3 protein: a member of the gap junction protein family
	*GNB4*	Guanine nucleotide-binding protein (G-protein) subunit beta 4; G proteins are involved as a modulator or transducer transmembrane signaling
	*NDRG1*	Belongs to the alpha/beta hydrolase superfamily: a cytoplasmic protein involved in stress responses, hormone responses, cell growth, and differentiation; is necessary for p53-mediated caspase activation and apoptosis
	*NGF*	Nerve Growth Factor: nerve growth stimulating activity
	*NTRK1*	High affinity nerve growth factor receptor tyrosine kinase: involved in the development and the maturation of the central and peripheral nervous systems
	*STING1* *(TMEM173)*	Regulator of the innate immune response to viral and bacterial infections
Gene expression and RNA processing	*ANG*	Angiogenin: a mediator of new blood vessel formation
	*ASCC1*	Subunit of the activating signal co-integrator 1 (ASC-1) complex: plays a role in DNA damage repair
	*DNMT1*	DNA (cytosine-5)-methyltransferase 1: methylates CpG residues
	*FUS*	Multifunctional protein involved in processes such as transcription regulation, RNA splicing, RNA transport, DNA repair and damage response; in neuronal cells, plays crucial roles in dendritic spine formation and stability, RNA transport, mRNA stability and synaptic homeostasis
	*HNRNPA1*	Involved in mRNA metabolism and transport
	*HOXD10*	Transcription factor which is part of a developmental regulatory system
	*IFRD1*	Protein related to interferon-gamma: this protein may function as a transcriptional co-activator/repressor that controls the growth and differentiation of specific cell types during embryonic development and tissue regeneration
	*LAS1L*	Involved in the biogenesis of the 60S ribosomal subunit
	*LITAF*	Plays a role in endosomal protein trafficking and in targeting proteins for lysosomal degradation
	*MED25*	Component of the transcriptional co-activator complex termed the Mediator complex, involved in the regulated transcription of nearly all RNA polymerase II-dependent genes
	*MORC2*	Essential for epigenetic silencing by the HUSH (human silencing hub) complex
	*PRDM12*	A transcriptional regulator of sensory neuronal specification that plays a critical role in pain perception
	*RBM7*	RNA-binding subunit of the trimeric nuclear exosome targeting (NEXT) complex, a complex that functions as an RNA exosome cofactor that directs a subset of non-coding short-lived RNAs for exosomal degradation
	*SOX10*	Transcription factor involved in developing and mature glia
	*TARDBP*	RNA-binding protein that is involved in various steps of RNA biogenesis and processing
	*TRIP4*	Transcription co-activator, which associates with transcriptional coactivators, nuclear receptors and basal transcription factors
	*ZNF106*	RNA-binding protein, required for normal expression and/or alternative splicing of a number of genes in the spinal cord and skeletal muscle
Transport	*ALS2*	Guanine nucleotide exchange factor for the small GTPase RAB5
	*BICD2*	A member of the Bicoid family: implicated in dynein-mediated motility along microtubules
	*DNM2*	Dynamin 2: microtubule-associated motor protein
	*FLVCR1*	Heme transporter that exports cytoplasmic heme
	*KIF1A*	Member of the kinesin family and functions as an anterograde motor protein
	*KIF1B*	A motor protein that transports mitochondria and synaptic vesicles
	*KIF5A*	A member of the kinesin family of proteins: microtubule-dependent motor
	*OPTN*	Optineurin: plays a role in the maintenance of the Golgi complex, in membrane trafficking and exocytosis
	*NIPA1*	Magnesium transporter
	*PLEKHG5*	Functions as a guanine exchange factor (GEF) for RAB26
	*SH3BP4*	Is involved in cargo-specific control of clathrin-mediated endocytosis, specifically controlling the internalization of a specific protein receptor
	*SLC5A7*	Sodium ion- and chloride ion-dependent high-affinity transporter that mediates choline uptake
	*SCN10A*	Tetrodotoxin-resistant voltage-gated sodium channel
	*SCN11A*	Voltage-gated sodium channel
	*SCN9A*	Voltage-dependent sodium channel
	*SLC12A6*	Potassium-chloride cotransporter
	*SLC5A2*	Sodium-dependent glucose transport protein
	*SLC5A3*	*Myo*-inositol transporter
	*SPG11*	Spatacsin: involved in the endolysosomal system and autophagy
	*SYT2*	Synaptic vesicle membrane protein: calcium sensor in vesicular trafficking and exocytosis
	*TFG*	Plays a role in the function of the endoplasmic reticulum (ER) and its associated microtubules
	*TRPA1*	Receptor-activated non-selective cation channel involved in pain detection and possibly also in cold perception, oxygen concentration perception, cough, itch, and inner ear function
	*TRPV4*	Non-selective calcium permeant cation channel involved in osmotic sensitivity and mechanosensitivity
	*TTR*	Transthyretin: one of the three prealbumins; is a carrier protein, which transports thyroid hormones in the plasma and cerebrospinal fluid; is involved in the transport of retinol in the plasma
Other	*LMNA*	The lamin family member: component of the nuclear lamina
	*PHYH*	Phytanoyl-CoA hydroxylase
	*NAGLU*	Alpha-N-acetylglucosaminidase: degrades heparan sulfate
	*TNNT2*	The tropomyosin-binding subunit of the troponin complex
	*TYMP*	An angiogenic factor which promotes angiogenesis and stimulates the *in vitro* growth of a variety of endothelial cells

List of genes was created based on [63]. Function of protein was described based on the GeneCard, UniProt, and OMIM databases [64,66,67].

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
