# Peer review of "A Yeast-Based Model for Hereditary Motor and Sensory Neuropathies: A Simple System for Complex, Heterogeneous Diseases"

_ijms, 2020, doi:10.3390/ijms21124277_

Round 1

Reviewer 1 Report

Rzepnikowska and coworkers wrote an interesting review on the potential use of yeast in hereditary motor sensory  neuropathies, mainly focused in Charcot-Marie-Tooth (CMT) disease. The work is a useful tool for researchers. Clearly the main advantage of the model is for screening drugs. To study the physiopathology of CMT in yeast Calcium handling or MAMs interaction, is not so clear probably a comment on this would be interesting. I strongly recommend English editing.

Author Response

We thank the Reviewer for his/her time an insightful comments that have made improvements to this manuscript. We have addressed all the points indicated by the Reviewer as follows.

  1. To study the physiopathology of CMT in yeast Calcium handling or MAMs interaction, is not so clear probably a comment on this would be interesting.

We have added the paragraph about study MAMs in yeast in the Section 4. Studies of CMT in yeast - based models for human genes with yeast orthologs [Lines 273-287] as follow: “The study of Mfn2 function is also of importance because mitofusins belong to the group of proteins important for the formation, regulation and function of endoplasmic reticulum (ER) and mitochondrial membrane contact sites (MCSs) called mitochondria-associated membranes (MAMs). MCSs are structures where membranes of different organelles are close and connected by a proteinaceous tether but do not fuse. The genes affecting homeostasis in MAMs are over-represented in the group of genes causing several hereditary neurodegenerative disorders such as Alzheimer’s disease [85-88] and Chorea-acanthocythosis [89]. This is because MCSs are involved in various processes including mitochondrial dynamics (fusion, fission), lipids metabolism, autophagy, cell survival, energy metabolism, calcium homeostasis and protein folding [90-92]. In the group of genes causing CMT disorders, besides Mfn2, there are other proteins with and without orthologs in yeast, which take part in processes at MAMs, such as VAPB, Opa1 or GDAP1 [93]. Studies on the function of one MCSs component may help uncover the role of MCSs in the development of several other neurodegenerative diseases. More details about the role of MAMs in neurodegeneration can be found in other reviews [85-88]“

  1. I strongly recommend English editing.

We have sent the manuscript to MDPI English editing service to improve English presentation (we have attached certificate).

Reviewer 2 Report

This manuscript described about hereditary motor sensory neuropathies. However, another review article includes the same theme of this article. For example, Biochimica et Biophysica Acta (BBA) - Volume 1852, Issue 4, April 2015, Pages 667-678, which is shown as a review: Hereditary motor and sensory neuropathies: Understanding molecular pathogenesis could lead to future treatment strategies. Author should describe the original aspect in the article.

Figure 1 and 2 are not clear. Especially, figure 1 has many information. It should be arranged to be easy to understand.

Author Response

We thank the Reviewer for his/her time an insightful comments that have made improvements to this manuscript. We have addressed all the points indicated by the Reviewer as follows.

  1. Moderate English changes required

The manuscript was edited by MDPI English editing service to improve English presentation (we have attached certificate).

  1. Another review article includes the same theme of this article. For example, Biochimica et Biophysica Acta (BBA) - Volume 1852, Issue 4, April 2015, Pages 667-678, which is shown as a review: Hereditary motor and sensory neuropathies: Understanding molecular pathogenesis could lead to future treatment strategies. Author should describe the original aspect in the article.

We have improve the Introduction to clarify the original aspect of our review [Lines 57-68]: “In this review, we raise the issue of using yeast as a model for studying neuropathies, in particular CMT disease, and present how it may help to overcome these three problematic, but ultimately basic, issues. Yeast systems offer many advantages that are still poorly utilized to investigate neuropathies in general. Many researchers do not realize that yeast may be a convenient model for studying ongoing processes in peripheral nerves diseases. We present the huge potential of this simple, unicellular organism to improve diagnostics, expand understanding of pathogenesis and accelerate the development of treatment. The studies of CMT disorder using a yeast model included here have not been summarized in any review to date; hence, we hope that by showing the broad spectrum of possibilities that yeast systems present, it may be more widely adopted as a useful tool in CMT research.”

We have added following sentence to the Section 3. Therapeutic approaches for Charcot-Marie-Tooth disorder [Lines 111-113] with suggested reference: “As the treatment strategies for different types of CMT were described in detail elsewhere [22].”

  1. Figure 1 and 2 are not clear. Especially, figure 1 has many information. It should be arranged to be easy to understand.

We have improved both Figure 1 and Figure 2. We have added, removed or rearranged some information. We hope that now both Figures present better and are easiest to understand.